# Zeroth-order (Non)-Convex Stochastic Optimization via Conditional Gradient and Gradient Updates

**Krishnakumar Balasubramanian**
Department of Statistics
University of California, Davis
kbala@ucdavis.edu

**Saeed Ghadimi** *
Department of Operations Research and Financial Engineering
Princeton University
sghadimi@princeton.edu

## Abstract

In this paper, we propose and analyze zeroth-order stochastic approximation algorithms for nonconvex and convex optimization. Specifically, we propose generalizations of the conditional gradient algorithm achieving rates similar to the standard stochastic gradient algorithm using only zeroth-order information. Furthermore, under a structural sparsity assumption, we first illustrate an implicit regularization phenomenon where the standard stochastic gradient algorithm with zeroth-order information adapts to the sparsity of the problem at hand by just varying the step-size. Next, we propose a truncated stochastic gradient algorithm with zeroth-order information, whose rate depends only poly-logarithmically on the dimensionality.

## 1 Introduction

In this work, we propose and analyze algorithms for solving the following stochastic optimization problem

$$\min_{x \in \mathcal{X}} \left\{ f(x) = \mathbf{E}_{\xi}[F(x,\xi)] = \int F(x,\xi)\,dP(\xi) \right\}, \tag{1.1}$$

where $\mathcal{X}$ is a closed convex subset of $\mathbb{R}^d$. The case of nonconvex objective function $f$ is ubiquitous in modern deep learning problems and developing provable algorithms for such problems has been a topic of intense research in the recent years [16, 11], along with the more standard convex case [1]. Several methods are available for solving such stochastic optimization problems under access to different oracle information, for example, function queries (zeroth-order oracle), gradient queries (first-order oracle), and higher-order oracles. In this work, we assume that we have access to noisy evaluation of $f$ through a stochastic zeroth-order oracle described in detail in Assumption 1. This oracle setting is motivated by several applications where only noisy function queries of problem (1.1) is available and obtaining higher-order information might not be possible. Such a situation occurs frequently for example, in simulation based modeling [29], selecting the tuning parameters of deep neural networks [32] and design of black-box attacks to deep networks [3]. It is worth noting that recently such zeroth-order optimization techniques have also been applied in the field of reinforcement learning [30, 4, 20]. Furthermore, methods using similar oracles have been studied in the literature under the name of derivative-free optimization [33, 5], bayesian optimization [21] and optimization with bandit feedback [2].

| Algorithm | Structure | Function Queries | References |
|---|---|---|---|
| ZSCG (Alg 1) | Nonconvex | $\mathcal{O}(d/\epsilon^4)$ | Theorem 2.1 |
| | Convex | $\mathcal{O}(d/\epsilon^3)$ | |
| Modified ZSCG (Alg 3) | Convex | $\mathcal{O}(d/\epsilon^2)$ | Theorem 2.2 |
| ZSGD (Alg 4) | Nonconvex, $s$-sparse | $\mathcal{O}\left((s\log d)^2/\epsilon^4\right)$ | Theorem 3.1 |
| Truncated ZSGD (Alg 5) | Convex, $s$-sparse | $\mathcal{O}\left(s(\log d/\epsilon)^2\right)$ | Theorem 3.2 |
| ZSGD | Convex | $\mathcal{O}(d/\epsilon^2)$ | [18, 7, 9] |
| | Nonconvex | $\mathcal{O}(d/\epsilon^4)$ | [9] |

Table 1: A list of complexity bounds for stochastic zeroth-order methods to find an $\epsilon$-stationary or $\epsilon$-optimal (see Definition 1.1) point of problem (1.1).

Algorithms available for solving problem (1.1) also depend crucially on the constraint set $\mathcal{X}$. First, consider the case of $\mathcal{X} = \mathbb{R}^d$. When first-order information is available, the rate of convergence of the standard Gradient Descent (GD) algorithm is dimension-independent [26]. Whereas when only the zeroth-order information is available, any algorithm (with estimated gradients) has (at least) linear dependence on $d$ [9, 18, 7]. This illustrates the main difference between the availability of different oracle information. Next, note that depending on the geometry of the constraint set $\mathcal{X}$, the cost of computing the projection to the set might be prohibitive. This lead to the re-emergence of Conditional Gradient (CG) algorithms recently [12, 15]. But the performance of the CG algorithm under the zeroth-order oracle is unexplored in the literature to the best of our knowledge, both under convex and nonconvex settings. Hence it is natural to ask if CG algorithms, with access to zeroth-order oracle has similar (or better) convergence rates compared to GD algorithms with zeroth-order information. We propose and analyze in Section 2 a classical version of CG algorithm with zeroth-order information and present convergence results. We then propose a modification in Section 2.2 that has improved rates, when $f$ is convex.

Notably, with zeroth-order information, the complexity of CG algorithms also depend linearly on the dimensionality, similar to the GD algorithms. We refer to this situation as the low-dimensional setting in the rest of the paper. This motivates us to examine assumptions under which one can achieve weaker dependence on the dimensionality while optimizing with zeroth-order information. In a recent work [34], the authors used a *functional sparsity* assumption, under which the function $f : \mathbb{R}^d \to \mathbb{R}$ to be optimized depends only on $s$ of the $d$ components, and proposed a LASSO based algorithm that has poly-logarithmic dependence on the dimensionality when $f$ is convex. We refer to this situation as the high-dimensional setting. In this work, we perform a refined analysis under a similar sparsity assumption for both convex and nonconvex objective functions. When the performance is measured by the size of the gradient, we show in Section 3 that zeroth-order GD algorithm (without using thresholding or LASSO approach of [34]), has poly-logarithmic dependence on the dimensionality thereby demonstrating an *implicit regularization* phenomenon in this setting. Note that this is applicable for both convex and nonconvex objectives. When the performance is measured by function values (as in the case of convex objective), we show that a simple thresholded zeroth-order GD algorithm achieves a poly-logarithmic dependence on dimensionality. This algorithm is notably less expensive than the algorithm proposed by [34].

**Our contributions:** To summarize the above discussion, in this paper we make the following contributions to the literature on zeroth-order stochastic optimization: (i) We first analyze a classical version of CG algorithm in the nonconvex (and convex) setting, under access to zeroth-order information and provide results on the convergence rates in the low-dimensional setting; (ii) We then propose and analyze a modified CG algorithm in the convex setting with zeroth-order information and show that it attains improved rates in the low-dimensional setting; (iii) Finally, we consider a zeroth-order stochastic gradient algorithm in the high-dimensional nonconvex (and convex) setting and illustrate an implicit regularization phenomenon. We also show that this algorithm achieves rates that depend only poly-logarithmically on dimensionality. Our contributions extend the applicability of zeroth-order stochastic optimization to the constrained and high-dimensional setting and also provide theoretical insights in the form of rates of convergence. A summary of the results is provided in Table 1.

## 1.1 Preliminaries

We now list the main assumptions we make in this work. Additional assumptions will be introduced in the appropriate sections as needed. We start with the assumption on the zeroth-order oracle.

**Assumption 1** *Let $\|\cdot\|$ be a norm on $\mathbb{R}^d$. For any $x \in \mathbb{R}^d$, the zeroth-order oracle outputs an estimator $F(x,\xi)$ of $f(x)$ such that $\mathbf{E}[F(x,\xi)] = f(x), \mathbf{E}[\nabla F(x,\xi)] = \nabla f(x), \mathbf{E}[\|\nabla F(x,\xi) - \nabla f(x)\|_*^2] \leq \sigma^2$, where $\|\cdot\|_*$ denotes the dual norm.*

It should be noted that in the above assumption, we do not observe $\nabla F(x,\xi)$ and we just assume that it is an unbiased estimator of gradient of $f$ and its variance is bounded. Furthermore, we make the following smoothing assumption about the noisy estimation of $f$.

**Assumption 2** *Function $F$ has Lipschitz continuous gradient with constant $L$, almost surely for any $\xi$, i.e., $\|\nabla F(y,\xi) - \nabla F(x,\xi)\|_* \leq L\|y - x\|$, which consequently implies that $|F(y,\xi) - F(x,\xi) - \langle \nabla F(x,\xi), y - x \rangle| \leq \frac{L}{2}\|y - x\|^2$.*

It is easy to see that the above two assumptions imply that $f$ also has Lipschitz continuous gradient with constant $L$ since

$$\|\nabla f(y) - \nabla f(x)\|_* \leq \mathbf{E}\left[\|\nabla F(y,\xi) - \nabla F(x,\xi)\|_*\right] \leq L\|y - x\| \tag{1.2}$$

due the Jensen's inequality for the dual norm. We now collect some facts about a gradient estimator based on the above zeroth-order information. Let $u \sim N(0, I_d)$ be a standard Gaussian random vector. For some $\nu \in (0, \infty)$ consider the smoothed function $f_\nu(x) = \mathbf{E}_u[f(x + \nu u)]$. Nesterov [27] has shown that $\nabla f_\nu(x) =$

$$\mathbf{E}_u\left[\frac{f(x + \nu u)}{\nu} u\right] = \mathbf{E}_u\left[\frac{f(x + \nu u) - f(x)}{\nu} u\right] = \frac{1}{(2\pi)^{d/2}} \int \frac{f(x + \nu u) - f(x)}{\nu} u\, e^{-\frac{\|u\|_2^2}{2}}\, du. \tag{1.3}$$

This relation implies that we can estimate gradient of $f_\nu$ by only using evaluations of $f$. In particular, one can define stochastic gradient of $f_\nu(x)$ as

$$G_\nu(x, \xi, u) = \frac{F(x + \nu u, \xi) - F(x, \xi)}{\nu} u, \tag{1.4}$$

which is an unbiased estimator of $\nabla f_\nu(x)$ under Assumption 1 since

$$\mathbf{E}_{u,\xi}[G_\nu(x, \xi, u)] = \mathbf{E}_u\left[\frac{f(x + \nu u) - f(x)}{\nu} u\right] = \nabla f_\nu(x).$$

We leverage some properties of $f_\nu$ due to Nesterov [27] in our proofs later, that we replicate in the supplementary material (Section A) for convenience. Finally, we define the following criterion which are used to analyze the complexity of our proposed algorithms.

**Definition 1.1** *Assume that a solution $\bar{x} \in \mathcal{X}$ as output of an algorithm and a target accuracy $\epsilon > 0$ are given. Then: (i) If $f$ is nonconvex, $\bar{x}$ is called an $\epsilon$-stationary point of the unconstrained variant of problem (1.1) if $\mathbf{E}[\|\nabla f(\bar{x})\|_*] \leq \epsilon$. For the constrained case, $\bar{x}$ should satisfies $\mathbf{E}[\langle \nabla f(\bar{x}), \bar{x} - u \rangle] \leq \epsilon$ for all $u \in \mathcal{X}$; (ii) If $f$ is convex, $\bar{x}$ is called an $\epsilon$-optimal point of problem (1.1) if $\mathbf{E}[f(\bar{x})] - f(x_*) \leq \epsilon$, where $x_*$ denotes an optimal solution of the problem.*

It should be pointed out that while the above performance measures are presented in expectation form, one can also use their high probability counterparts. Since, convergence results in this case can be obtained by making sub-Gaussian tail assumptions on the output of the zeroth-order oracle and using the standard two-stage process presented in [9, 19], we do not elaborate more on this approach. Furthermore, note that the aforementioned measures for evaluating the algorithms are from the derivative-free optimization point of view. In the literature on optimization with bandit feedback, the preferred performance measure is the so-called regret of the algorithm [2, 31] which may have a different behavior than our performance measures.

## 2 Zeroth-order Stochastic Conditional Gradient Type Method

In this section, we study zeroth-order stochastic conditional gradient (ZSCG) algorithms in the low-dimensional setting for solving constrained stochastic optimization problems. In particular, we incorporate a variant of the gradient estimate defined in (1.4) into the framework of the classical CG method and provide its convergence analysis in Subsection 2.1. We also present improved rates for a variant of this method in Subsection 2.2 when $f$ is convex. Throughout this section, we assume that $\mathbb{R}^d$ is equipped with the self-dual Euclidean norm i.e., $\|\cdot\| = \|\cdot\|_2$. We also make the following natural boundedness assumption.

---
**Algorithm 1** Zeroth-order Stochastic Conditional Gradient Method
---
Input: $z_0 \in \mathcal{X}$, smoothing parameter $\nu > 0$, non-negative sequence $\alpha_k$, positive integer sequence $m_k$, iteration limit $N \geq 1$ and probability distribution $P_R(\cdot)$ over $\{1, \ldots, N\}$.

**for** $k = 1, \ldots, N$ **do**

    1. Generate $u_k = [u_{k,1}, \ldots, u_{k,m_k}]$, where $u_{k,j} \sim N(0, I_d)$, call the stochastic oracle to compute $m_k$ stochastic gradient $G_\nu^{k,j}$ according to (1.4) and take their average:

$$\bar{G}_\nu^k \equiv \bar{G}_\nu(z_{k-1}, \xi_k, u_k) = \frac{1}{m_k} \sum_{j=1}^{m_k} \frac{F(z_{k-1} + \nu u_{k,j}, \xi_{k,j}) - F(z_{k-1}, \xi_{k,j})}{\nu} u_{k,j}. \qquad (2.1)$$

    2. Compute

$$x_k = \operatorname*{argmin}_{u \in \mathcal{X}} \langle \bar{G}_\nu^k, u \rangle, \qquad (2.2)$$

$$z_k = (1 - \alpha_k) z_{k-1} + \alpha_k x_k. \qquad (2.3)$$

**end for**

Output: Generate $R$ according to $P_R(\cdot)$ and output $z_R$.

---

**Assumption 3** *The feasible set $\mathcal{X}$ is bounded such that $\max_{x,y \in \mathcal{X}} \|y - x\| \leq D_\mathcal{X}$ for some $D_\mathcal{X} > 0$. Moreover, for all $x \in \mathcal{X}$, there exists a constant $B > 0$ such that $\|\nabla f(x)\| \leq B$.*

We should point out that under Assumptions 1 and 2, the second statement in Assumption 3 follows immediately by the first one and choosing $B := L D_\mathcal{X} + \|\nabla f(x_*)\|$. However, we just use $B$ in our analysis for simplicity.

## 2.1 Zeroth-order Stochastic Conditional Gradient Method

The vanilla ZSCG method is formally presented in Algorithm 1 and a few remarks about it follows. First, note that this algorithm differs from the classical CG method in estimating the gradient using zeroth-order information and in outputting a random solution from the generated trajectory. This randomization scheme is the current practice in the literature to provide convergence results for nonconvex stochastic optimization (see e.g., [9, 28]). Second, $\bar{G}_\nu^k$ is the averaged variant of the gradient estimator presented in Subsection 1.1 and is still an unbiased estimator of $\nabla f_\nu(z_{k-1})$. Moreover, it can be easily seen that it has a reduced variance with respect to the individual estimators i.e.,

$$\mathbf{E}[\|\bar{G}_\nu^k - \nabla f_\nu(z_{k-1})\|^2] \leq \frac{1}{m_k} \mathbf{E}[\|G_\nu^{k,j} - \nabla f_\nu(z_{k-1})\|^2]. \qquad (2.4)$$

We emphasize that the use of the above variance reduction technique in stochastic CG methods is standard and has been previously proposed and leveraged in several works (see e.g., [19, 13, 28, 22, 23, 10]). Indeed, when exact gradient is not available, an error term appears in the convergence analysis which should converge to 0 at a certain rate as the algorithm moves forward. Hence, the choice of $m_k$ plays a key role in the convergence analysis of Algorithm 1. $\bar{G}_\nu^k$ can be also viewed as a biased estimator for $\nabla f(z_{k-1})$. Finally, since $f$ is possibly nonconvex, we need a different criteria than the optimality gap to provide convergence analysis of Algorithm 1. The well-known Frank-Wolfe Gap given by

$$g_\mathcal{X}^k \equiv g_\mathcal{X}(z_{k-1}) := \langle \nabla f(z_{k-1}), z_{k-1} - \hat{x}_k \rangle, \quad \text{where} \quad \hat{x}_k = \operatorname*{argmin}_{u \in \mathcal{X}} \langle \nabla f(z_{k-1}), u \rangle, \qquad (2.5)$$

has been widely use in the literature to show rate of convergence of the CG methods when $f$ is convex (see e.g., [8, 6, 14]). In this case, it is easy to see that

$$f(z_{k-1}) - f^* \leq g_\mathcal{X}(z_{k-1}). \qquad (2.6)$$

When $f$ is nonconvex, this criteria is still useful since $\langle \nabla f(z_{k-1}), z_{k-1} - u \rangle \leq g_\mathcal{X}(z_{k-1})$, $\forall u \in \mathcal{X}$, which implies that one can obtain an approximate stationary point of problem (1.1) by minimizing $g_\mathcal{X}^k$, in the view of Definition 1.1. Note that in our setting, this quantity is not exactly computable and it is only used to provide convergence analysis of Algorithm 1 as shown in the next result.

**Theorem 2.1** *Let $\{z_k\}_{k \geq 0}$ be generated by Algorithm 1 and Assumptions 1, 2, and 3 hold.*

1. *Let $f$ be nonconvex, bounded from below by $f^*$, and let the parameters of the algorithm be set as*

$$\nu = \sqrt{\frac{2B_{L\sigma}}{N(d+3)^3}}, \quad \alpha_k = \frac{1}{\sqrt{N}}, \quad m_k = 2B_{L\sigma}(d+5)N, \quad \forall k \geq 1 \qquad (2.7)$$

*for some constant $B_{L\sigma} \geq \max\{\sqrt{B^2 + \sigma^2}/L, 1\}$ and a given iteration bound $N \geq 1$. Then we have*

$$\mathbf{E}[g_{\mathcal{X}}^R] \leq \frac{f(z_0) - f^* + LD_{\mathcal{X}}^2 + 2\sqrt{B^2 + \sigma^2}}{\sqrt{N}}, \qquad (2.8)$$

*where $R$ is uniformly distributed over $\{1, \dots, N\}$ and $g_k$ is defined in (2.5). Hence, the total number of calls to the zeroth-order stochastic oracle and linear subproblems required to be solved to find an $\epsilon$-stationary point of problem (1.1) are, respectively, bounded by*

$$\mathcal{O}\left(\frac{d}{\epsilon^4}\right), \quad \mathcal{O}\left(\frac{1}{\epsilon^2}\right). \qquad (2.9)$$

2. *Let $f$ be convex and let the parameters be set to*

$$\nu = \sqrt{\frac{2B_{L\sigma}}{N^2(d+3)^3}}, \quad \alpha_k = \frac{6}{k+5}, \quad m_k = 2B_{L\sigma}(d+5)N^2, \quad \forall k \geq 1. \qquad (2.10)$$

*Then we have*

$$\mathbf{E}[f(z_N)] - f^* + \mathbf{E}[g_{\mathcal{X}}^R] \leq \frac{120[f(z_0) - f(x_*)]}{(N+3)^3} + \frac{36LD_{\mathcal{X}}^2}{N+5} + \frac{\sqrt{B^2 + \sigma^2}}{N} \qquad (2.11)$$

*where $R$ is random variable from $\{1, \dots, N\}$ whose probability distribution is given by*

$$P_R(R = k) = \frac{\alpha_k \Gamma_N}{2\Gamma_N(1 - \Gamma_N)}, \qquad \Gamma_k = \prod_{i=1}^{k}\left(1 - \frac{\alpha_i}{2}\right), \quad \Gamma_0 = 1. \qquad (2.12)$$

*Hence, the total number of calls to the zeroth-order stochastic oracle and linear subproblems required to be solved to find and $\epsilon$-optimal solution of problem (1.1) are, respectively, bounded by*

$$\mathcal{O}\left(\frac{d}{\epsilon^3}\right), \quad \mathcal{O}\left(\frac{1}{\epsilon}\right). \qquad (2.13)$$

**Remark 1** *Observe that the complexity bounds in (2.9), in terms of $\epsilon$, match the ones obtained in [10, 28, 23] for stochastic CG method with first-order oracle applied to nonconvex problems. For convex problems, similar observation can be made for terms in (2.13) which match the ones in [13, 10]. Note that the linear dependence of our complexity bounds on $d$ is unimprovable due to the lower bounds for zeorth-order algorithms applied to convex optimization problems [7]. We conjecture that this is also the case for nonconvex problems.* ∎

## 2.2 Improved Rates for Convex Problems

Our goal in this subsection is to improve the complexity bounds of the ZCSG method when $f$ is convex. Recall that the ZSCG method presented in Section 2.1 involves two main steps: the gradient evaluation step and the linear optimization step. Motivated by [19], we now propose a modified algorithm that allows one to skip the gradient evaluation from time to time. Notice that, as our gradients are estimated by calling the zeroth-order oracle, this directly reduces the number of calls to the zeroth-order oracle. We first state a subroutine in Algorithm 2 used in our modified algorithm. Note that Algorithm 2 is indeed the zeroth-order conditional gradient method for inexactly solving the following quadratic program

$$P_{\mathcal{X}}(x, g, \gamma) = \underset{u \in \mathcal{X}}{\operatorname{argmin}}\left\{\langle g, u \rangle + \frac{\gamma}{2}\|u - x\|^2\right\}, \qquad (2.15)$$

which is the standard subproblem of stochastic first-order methods applied to a minimization problem when $g$ is an unbiased stochastic gradient of the objective function at $x$. We now present Algorithm 3 which applies the CG method to inexactly solve subproblems of the stochastic accelerated gradient

---

**Algorithm 2** Inexact Conditional Gradient (ICG) method

---

Input: $(x, g, \gamma, \mu)$.
Set $\bar{y}_0 = x$, $t = 1$, and $\kappa = 0$..
**while** $\kappa = 0$ **do**

$$y_t = \operatorname*{argmin}_{u \in \mathcal{X}} \{h_\gamma(u) := \langle g + \gamma(\bar{y}_{t-1} - x), u - \bar{y}_{t-1}\rangle\} \qquad (2.14)$$

If $h_\gamma(y_t) \geq -\mu$, set $\kappa = 1$.
Else $\bar{y}_t = \frac{t-1}{t+1}\bar{y}_{t-1} + \frac{2}{t+1}y_t$ and $t = t+1$.
**end while**
Output $\bar{y}_t$.

---

method. This way of using CG methods can significantly improve the total number of calls to the stochastic oracle. Our next result provides convergence analysis of this algorithm.

---

**Algorithm 3** Zeroth-order Stochastic Accelerated Gradient Method with Inexact Updates

---

Input: $z_0 = x_0 \in \mathcal{X}$, smoothing parameter $\nu > 0$, sequences $\alpha_k$, $m_k$, $\gamma_k$, $\mu_k$, and iteration limit $N \geq 1$.
**for** $k = 1, \ldots, N$ **do**
    1. Set

$$w_k = (1 - \alpha_k)z_{k-1} + \alpha_k x_{k-1} \qquad (2.16)$$

    2. Generate $u_k = [u_{k,1}, \ldots, u_{k,m_k}]$, where $u_{k,j} \sim N(0, I_d)$, call the stochastic oracle $m_k$ times to compute $\bar{G}_\nu^k \equiv \bar{G}_\nu(w_k, \xi_k, u_k)$ as given by (2.1), and set

$$x_k = ICG(x_{k-1}, \bar{G}_\nu^k, \gamma_k, \mu_k), \qquad (2.17)$$

where $ICG(\cdot)$ is the output of Algorithm 2 with input $(x_{k-1}, \bar{G}_\nu^k, \gamma_k)$.
    3. Set

$$z_k = (1 - \alpha_k)z_{k-1} + \alpha_k x_k \qquad (2.18)$$

**end for**
Output: $z_N$

---

**Theorem 2.2** *Let $\{z_k\}_{k \geq 1}$ be generated by Algorithm 3, the function $f$ be convex, and*

$$\alpha_k = \frac{2}{k+1}, \quad \gamma_k = \frac{4L}{k}, \quad \mu_k = \frac{LD_X^0}{kN}, \quad \nu = \frac{1}{\sqrt{2N}}\max\left\{\frac{1}{d+3}, \sqrt{\frac{D_X^0}{d(N+1)}}\right\}$$

$$m_k = \frac{k(k+1)}{D_X^0}\max\{(d+5)B_{L\sigma}N, d+3\}, \quad \forall k \geq 1, \qquad (2.19)$$

*and for some constants $D_X^0 \geq \|x_0 - x_*\|^2$ and $B_{L\sigma} \geq \max\{\sqrt{B^2 + \sigma^2}/L, 1\}$. Then under Assumptions 1, 2, and 3, we have*

$$\mathbf{E}[f(z_N) - f(x_*)] \leq \frac{12LD_X^0}{N(N+1)}. \qquad (2.20)$$

*Hence, the total number of calls to the stochastic oracle and linear subproblems solved to find and $\epsilon$-stationary point of problem (1.1) are, respectively, bounded by*

$$\mathcal{O}\left(\frac{d}{\epsilon^2}\right), \quad \mathcal{O}\left(\frac{1}{\epsilon}\right). \qquad (2.21)$$

**Remark 2** *Observe that while the number of linear subproblems required to find an $\epsilon$-optimal solution of problem (1.1) is the same for both Algorithms 1 and 3, the number of calls to the stochastic zeroth-order oracle in Algorithm 3 is significantly smaller than that of Algorithm 1. It is also natural to ask if such an improvement is achievable when $f$ is nonconvex. This situation is more subtle and the answer depends on the performance measure used to measure the rate of convergence. Indeed, we can obtain improved complexity bounds for a different performance measure than the Frank-Wolfe*

**Algorithm 4** Zeroth-Order Stochastic Gradient Method

---

Input: $x_0 \in \mathbb{R}^d$, smoothing parameter $\nu > 0$, iteration limit $N \geq 1$, a probability distribution $P_R$ supported on $\{0, \ldots, N-1\}$.
**for** k =1, ..., N **do**
    Generate $u_k \sim N(0, I_d)$, call the stochastic oracle, and compute $G_\nu(x_{k-1}, \xi_k, u_k)$ as defined in (1.4) and set $x_k = x_{k-1} - \gamma_k G_\nu(x_{k-1}, \xi_k; u_k)$.
**end for**
Output: Generate $R$ according to $P_R(\cdot)$ and output $x_R$.

---

*gap with a modified algorithm. However, the complexity bounds are of the same order as (2.9) in terms of the Frank-Wolfe gap for the modified algorithm. For the sake of completeness, we add this algorithm and its convergence analysis in the supplementary material in Section D.* ∎

## 3 Zeroth-order Stochastic Gradient Methods

In this section, we study unconstrained variant of problem 1.1 i.e, $\mathcal{X} = \mathbb{R}^d$, under certain sparsity assumptions on the objective function $f$ to facilitate zeroth-order optimization in high-dimensions. Recently, [34] considered the convex case and proposed algorithms for high-dimensional zeroth-order stochastic optimization. Motivated by [34], we make the following assumption.

**Assumption 4** *For any $x \in \mathbb{R}^d$, we have $\|\nabla f(x)\|_0 \leq s$, i.e., the gradient is $s$-sparse, where $s \ll d$.*

Note that the above assumption implies $\|\nabla f(x)\|_2 \leq \sqrt{s}\|\nabla f(x)\|_\infty$ and $\|\nabla f(x)\|_1 \leq s\|\nabla f(x)\|_\infty$, for all $x \in \mathbb{R}^d$. Furthermore, this assumption also implies that $\|\nabla f_\nu(x)\|_0 \leq s$ for all $x \in \mathbb{R}^d$ since $\nabla f_\nu(x) = \mathbf{E}_u[\nabla f(x + \nu u)]$. To exploit the above sparsity assumption, we assume that the primal space $\mathbb{R}^d$ is equipped with the $l_\infty$ norm throughout this section. More specifically, we assume that Assumptions 1 and 2 hold with the choice of $\|\cdot\| = \|\cdot\|_\infty$ and its dual norm $\|\cdot\|_* = \|\cdot\|_1$. We now present zeroth-order stochastic gradient methods for solving problem (1.1) when $f$ is nonconvex and convex, in Subsections 3.1 and 3.2 respectively.

### 3.1 Zeroth-order Stochastic Gradient Method for Nonconvex Problems

In this subsection, we consider the zeroth-order stochastic gradient method presented in [9] (provided in Algorithm 4 for convenience) and provide a refined convergence analysis for it under the sparsity assumption 1, when $f$ is nonconvex. Our main convergence result for Algorithm 4 under the gradient sparsity assumption is stated below.

**Theorem 3.1** *Let $\{x_k\}_{k \geq 0}$ be generated by Algorithm 4 and stepsizes are chosen such that $\forall k \geq 1$,*

$$\gamma_k = \frac{1}{2L\hat{C} \log d} \min \left\{ \frac{1}{12\hat{s} \log d}, \sqrt{\frac{D_0 L\hat{C}}{2N\sigma^2}} \right\}, \quad \nu \leq \frac{1}{\sqrt{L\hat{C} \log d}} \min \left\{ \sqrt{\frac{2\sigma^2}{L}}, \sqrt{\frac{D_0}{N}} \right\} \quad (3.1)$$

*for some $\hat{s} \geq s$, $\hat{C} \geq C$ (the universal constant defined in Lemma C.1), and $D_0 \geq f(x_0) - f^*$. Assume that $f$ is nonconvex. Then under Assumptions 1, 2, and 4, we have*

$$\mathbf{E}_\zeta \left[ \|\nabla f(x_R)\|_1^2 \right] \leq \frac{150 L\hat{C} D_0 \hat{s} s (\log d)^2}{N} + \frac{54\sigma \sqrt{2L\hat{C} D_0} \, s \log d}{\sqrt{N}}, \quad (3.2)$$

*where $\zeta = \{\xi, u, R\}$ and $R$ is uniformly distributed over $\{0, \ldots, N-1\}$. Hence, the total number of calls to the stochastic oracle (number of iterations) required to find an $\epsilon$-stationary point of problem (1.1), in the view of Definition 1.1, is bounded by*

$$\mathcal{O}\left( \frac{(\hat{s} \log d)^2}{\epsilon^4} \right). \quad (3.3)$$

**Remark 3** *Note that the above theorem establishes rate of convergence of Algorithm 4 which only poly-logarithmically depends on the problem dimension $d$, by just selecting the step-size appropriately, under additional assumption that the gradient is sparse. This significantly improves the linear dimensionality dependence of the rate of convergence of this algorithm as presented in [9] for general nonconvex smooth problems.* ∎

---
**Algorithm 5** Truncated Zeroth-Order Stochastic Gradient Method
---
Given a positive integer $\hat{s}$, replace updating step of Algorithm 4 with

$$x_k = P_{\hat{s}} \left( x_{k-1} - \gamma_k G_\nu(x_{k-1}, \xi_k; u_k) \right), \tag{3.4}$$

where $P_{\hat{s}}(x)$ keeps the top $\hat{s}$ largest absolute value of components of $x$ and make the others $0$.

---

**Remark 4** *Remarkably, Algorithm 4 does not require any special operation to adapt to the sparsity assumption. This demonstrates an* implicit regularization *phenomenon exhibited by the zeroth-order stochastic gradient method in the high-dimensional setting when the performance is measured by the size of the gradient in the dual norm. We emphasize that the choice of the performance measure is motivated by the fact that we allow $f$ to be nonconvex. Trivially, the result also applies to the case when $f$ is convex, for the same performance measure.* ∎

### 3.2 Zeroth-order Stochastic Gradient Method for Convex Problems

We now consider the case when the function $f$ is convex. In this setting, a more natural performance measure is the convergence of optimality gap in terms of the function values. For this situation, we propose and analyze a truncate variant of Algorithm 4 that demonstrates similar poly-logarithmic dependence on the dimensionality. To proceed, in addition to Assumption 4, we also make the following sparsity assumption on the optimal solution of problem (1.1).

**Assumption 5** *Problem (1.1) has a sparse optimal solution $x_*$ such that $\|x_*\|_0 \leq s^*$, where $s^* \approx s$.*

Our algorithm for the convex setting is presented in Algorithm 5. Note that this algorithm could be considered as a truncated variant of Algorithm 4 and a zeroth-order stochastic variant of the truncated gradient descent algorithm [17]. In the next result, we present convergence analysis of this algorithm.

**Theorem 3.2** *Let $\{x_k\}_{k \geq 1}$ be generated by Algorithm 4, $f$ is convex, Assumptions 1, 2, 4, and 5 hold. Also assume the stepsizes are chosen such that, $\forall k \geq 1$,*

$$\gamma_k = \frac{1}{4\hat{C}\hat{s}\log d} \min \left\{ \frac{1}{12L\hat{s}\log d}, \sqrt{\frac{D_X^0 \hat{C}\hat{s}}{3N\sigma^2}} \right\}, \quad \nu \leq \sqrt{\log d} \min \left\{ \frac{\sigma}{\log d}, \sqrt{\frac{\hat{s}^2 D_X^0}{N}} \right\} \tag{3.5}$$

*for some $\hat{C} \geq C$, $\hat{s} \geq \max\{s, s^*\}$, and $D_X^0 \geq \|x_0 - x_*\|^2$.*

$$\mathbf{E}\left[f(\bar{x}_N) - f^*\right] \leq \frac{52L\hat{C}D_X^0 \hat{s}^2 (\log d)^2}{N} + \frac{69\sigma\sqrt{3\hat{C}D_X^0 \hat{s}} \log d}{\sqrt{N}}, \tag{3.6}$$

*where $\bar{x}_N = \frac{\sum_{k=0}^{N-1} x_k}{N}$. Hence, the total number of calls to the stochastic oracle (number of iterations) required to find an $\epsilon$-optimal point of problem (1.1) is bounded by*

$$\mathcal{O}\left( \hat{s} \left( \frac{\log d}{\epsilon} \right)^2 \right). \tag{3.7}$$

**Remark 5** *While for convex case, similar to the nonconvex case, the complexity of Algorithm 5 depends poly-logarithmically on $d$, it only linearly depends on the choice of $\hat{s}$, facilitating zeroth-order stochastic optimization in high-dimensions under sparsity assumptions.* ∎

**Remark 6** *As discussed in detail in [34], both Assumption 4 and 5 are implied when we assume the function $f$ depends on only $s$ of the $d$ coordinates. But, both Assumption 4 and 5 are comparatively weaker than that assumption. Furthermore, unlike [34], we do not make any assumption on the sparsity or smoothness of the second-order derivative of the objective function $f$ for our results.* ∎

**Remark 7** *As mentioned before, [34] considers only the convex case. Furthermore, their gradient estimator with zeroth-order oracle requires $poly(s, s^*, \log d)$ function queries in each iteration whereas our estimator is based on only one function query per iteration. Moreover, [34] requires computationally expensive debiased Lasso estimators whereas our method requires only simple thresholding operations (for convex case) to handle sparsity.* ∎

## 4   Future Work

Two concrete extensions are possible for future work. First, for our results, we focus on performance measures common in the optimization setting. It is interesting to extend our results to the bandit setting, where the performance is measured via regret of the algorithm. Next, the performance of conditional gradient algorithm in the high-dimensional constrained optimization setting is not well-explored; the interaction between the geometry of the constraint set, sparsity structure and zeroth-order information is extremely interesting to explore. Finally, lower bounds can be explored for the cases considered in this paper when $f$ is nonconvex.

## Footnotes

*Both authors contributed equally and are listed in alphabetical order.

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
