[Supplementary Material]

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

## A  Relevant Results from [27]

In this section, for completeness, we replicate relevant results from [27], that are required for our proofs.

**Theorem A.1** *The following statements hold for any function $f$ whose gradient is Lipschitz continuous with constant $L$.*

    *a) The gradient of $f_\nu$ is Lipschitz continuous with constant $L_\nu$ such that $L_\nu \leq L$.*

    *b) For any $x \in \mathbb{R}^d$,*

$$|f_\nu(x) - f(x)| \quad \leq \quad \frac{\nu^2}{2}Ld, \tag{A.8}$$

$$\|\nabla f_\nu(x) - \nabla f(x)\| \quad \leq \quad \frac{\nu}{2}L(d+3)^{\frac{3}{2}}. \tag{A.9}$$

    *c) For any $x \in \mathbb{R}^n$,*

$$\frac{1}{\nu^2}\mathbf{E}_u[\{f(x+\nu u) - f(x)\}^2\|u\|^2] \leq \frac{\nu^2}{2}L^2(d+6)^3 + 2(d+4)\|\nabla f(x)\|^2. \tag{A.10}$$

## B  Proofs for Section 2

We present all the proofs for Section 2 below. Recall that, we assumed that $\|\cdot\| = \|\cdot\|_2$ in Section 2. In order to prove Theorem 2.1, we need the following result that provides upper bounds for the variance of our gradient estimator.

**Lemma B.1** *Let $\bar{G}_\nu^k$ be computed by (2.1). Then under Assumptions 1, 2 and 3, we have*

$$\mathbf{E}[\|\bar{G}_\nu^k - \nabla f_\nu(z_{k-1})\|^2] \quad \leq \quad \frac{2(d+5)(B^2+\sigma^2)}{m_k} + \frac{\nu^2}{2m_k}L^2(d+3)^3, \tag{B.11}$$

$$\mathbf{E}[\|\bar{G}_\nu^k - \nabla f(z_{k-1})\|^2] \quad \leq \quad \frac{4(d+5)(B^2+\sigma^2)}{m_k} + \frac{3\nu^2}{2}L^2(d+3)^3. \tag{B.12}$$

*Proof.* First note that using (A.10) for function $F$ instead of $f$, under Assumptions 1 and 2, we obtain

$$\mathbf{E}[\|G_\nu^{k,j}\|^2] \leq \frac{\nu^2 L^2}{2}\mathbf{E}\left[\|u\|^6\right] + 2\mathbf{E}_\xi[\|\nabla F(z_{k-1}, \xi_k)\|^2]\mathbf{E}_u\left[\|u\|^4\right]$$
$$\leq \frac{\nu^2 L^2}{2}(d+6)^3 + 2\left[\|\nabla f(z_{k-1})\|^2 + \sigma^2\right](d+4),$$

where the second inequality follows from the fact that $\mathbf{E}[\|u\|^k] \leq (d+k)^{k/2}$ for any $k \geq 2$ due to Nesterov [27]. Also noting (1.4), (2.4), and the fact that $\|\nabla f_\nu\| \leq B$ under Assumption 3, we have

$$\mathbf{E}[\|\bar{G}_\nu^k - \nabla f_\nu(z_{k-1})\|^2] \leq \frac{1}{m_k}\left(\mathbf{E}[\|G_\nu^{k,j}\|^2] + B^2\right),$$

which together with the above relation clearly imply (B.11). We can then obtain (B.12) by noting (A.9) and the fact that

$$\mathbf{E}[\|\bar{G}_\nu^k - \nabla f(z_{k-1})\|^2] \leq 2\mathbf{E}[\|\bar{G}_\nu^k - \nabla f_\nu(z_{k-1})\|^2] + 2\mathbf{E}[\|\nabla f_\nu(z_{k-1}) - \nabla f(z_{k-1})\|^2].$$

∎

### B.1  Proof of Theorem 2.1

*Proof.* Denoting $\Delta_k = \bar{G}_\nu^k - \nabla f(z_{k-1})$, noting (1.2), (2.3), and (2.5), we have

$$f(z_k) \leq f(z_{k-1}) + \langle \nabla f(z_{k-1}), z_k - z_{k-1}\rangle + \frac{L}{2}\|z_k - z_{k-1}\|^2$$

$$= f(z_{k-1}) + \alpha_k\langle \nabla f(z_{k-1}), x_k - z_{k-1}\rangle + \frac{L\alpha_k^2}{2}\|x_k - z_{k-1}\|^2$$

$$\leq f(z_{k-1}) + \alpha_k\langle \nabla f(z_{k-1}), \hat{x}_k - z_{k-1}\rangle + \frac{L\alpha_k^2}{2}\left[\|x_k - z_{k-1}\|^2 + \|x_k - \hat{x}_k\|^2\right] + \frac{\|\Delta_k\|^2}{2L}$$

$$\leq f(z_{k-1}) - \alpha_k g_\mathcal{X}^k + LD_\mathcal{X}^2\alpha_k^2 + \frac{\|\Delta_k\|^2}{2L}, \tag{B.13}$$

where the last inequality follows from boundedness of the feasible set, (2.5), and the fact that

$$\langle \nabla f(z_{k-1}) + \Delta_k, x_k - u \rangle \leq 0 \ \ \forall u \in \mathcal{X}$$

due to the optimality condition of (2.2). Taking expectation from both sides of the above inequality, summing them up, rearranging the terms, and noting Lemma B.1, we obtain

$$\sum_{k=1}^{N} \alpha_k \mathbf{E}[g_{\mathcal{X}}^k] \leq f(z_0) - f^* + LD_{\mathcal{X}}^2 \sum_{k=1}^{N} \alpha_k^2 + \frac{\nu^2}{2} LN(d+3)^3 + \frac{2(d+5)(B^2+\sigma^2)}{L} \sum_{k=1}^{N} \frac{1}{m_k}.$$

Hence, choosing $\alpha_k = \alpha_1$ and $m_k = m_1$ for all $k \geq 1$, and noting that $R$ is a uniform random variable, we have

$$\mathbf{E}[g_{\mathcal{X}}^R] = \frac{\sum_{k=1}^{N} \mathbf{E}[g_{\mathcal{X}}^k]}{N} = \frac{\sum_{k=1}^{N} \alpha_k \mathbf{E}[g_{\mathcal{X}}^k]}{\sum_{k=1}^{N} \alpha_k} \leq \frac{f(z_0) - f^*}{N\alpha_1} + LD_{\mathcal{X}}^2 \alpha_1 + \frac{\nu^2}{2\alpha_1} L(d+3)^3$$

$$+ \frac{2(d+5)(B^2+\sigma^2)}{L\alpha_1 m_1},$$

which together with (2.7) imply (2.8). Hence, (2.9) follows by noting that the total number of calls to the stochastic oracle is bounded by $\sum_{k=1}^{N} m_k$.

Now assume that $f$ is convex. Hence, by (2.6) and (B.13), we have

$$f(z_k) - f(x_*) \leq (1 - \tfrac{\alpha_k}{2})(f(z_{k-1}) - f(x_*)) - \frac{\alpha_k g_{\mathcal{X}}^k}{2} + LD_{\mathcal{X}}^2 \alpha_k^2 + \frac{\|\Delta_k\|^2}{2L}$$

Taking expectation from both sides of the above inequality, dividing them by $T_k$, and summing them up, and noting (2.12), we obtain

$$\frac{\mathbf{E}[f(z_N)] - f^*}{\Gamma_N} + \sum_{k=1}^{N} \frac{\alpha_k \mathbf{E}[g_{\mathcal{X}}^k]}{2\Gamma_k} \leq f(z_0) - f^* + LD_{\mathcal{X}}^2 \sum_{k=1}^{N} \frac{\alpha_k^2}{\Gamma_k} + \frac{1}{2L} \sum_{k=1}^{N} \frac{\mathbf{E}\left[\|\Delta_k\|^2\right]}{\Gamma_k},$$

which together with the fact that

$$\sum_{k=1}^{N} \frac{\alpha_k}{2\Gamma_k} = \frac{1 - \Gamma_N}{\Gamma_N}, \ \ 1 - \Gamma_1 \leq 1 - \Gamma_N \leq 1$$

due to (2.12), imply that

$$\mathbf{E}[f(z_N)] - f^* + \mathbf{E}[g_{\mathcal{X}}^R] \leq \frac{\Gamma_N}{1 - \Gamma_N} \left[ f(z_0) - f^* + LD_{\mathcal{X}}^2 \sum_{k=1}^{N} \frac{\alpha_k^2}{\Gamma_k} + \frac{2(d+5)(B^2+\sigma^2)}{L} \sum_{k=1}^{N} \frac{1}{\Gamma_k m_k} \right.$$

$$\left. + \frac{\nu^2}{2} L(d+3)^3 \sum_{k=1}^{N} \frac{1}{\Gamma_k} \right] \tag{B.14}$$

Now noting (2.10) and (2.12), we have

$$\Gamma_k = \frac{60}{(k+3)(k+4)(k+5)}, \qquad \sum_{k=1}^{N} \frac{\alpha_k^2}{\Gamma_k} \leq \sum_{k=1}^{N} \frac{3(k+3)}{5} = \frac{3N(N+7)}{10},$$

$$\Gamma_N \sum_{k=1}^{N} \frac{1}{\Gamma_k m_k} \leq \frac{1}{4(d+5)B_{L\sigma}N}, \qquad \Gamma_N \sum_{k=1}^{N} \frac{1}{\Gamma_k} \leq N.$$

Combining the above relations, we get (2.11) and (2.13). ∎

## B.2  Proof of Theorem 2.2

*Proof.* First, note that by (1.2), we have

$$
\begin{aligned}
f_\nu(z_k) \ &\leq \ f_\nu(w_k) + \langle \nabla f_\nu(w_k), z_k - w_k \rangle + \frac{L}{2} \|z_k - w_k\|^2 \\
&\leq \ (1 - \alpha_k) f_\nu(z_{k-1}) + \alpha_k \left[ f_\nu(w_k) + \langle \nabla f_\nu(w_k), x_k - w_k \rangle \right] \\
&+ \ \frac{L\alpha_k^2}{2} \|x_k - x_{k-1}\|^2,
\end{aligned}
\tag{B.15}
$$

where the second inequality follows from convexity of $f_\nu$, (2.16), and (2.18). Also note that by (2.14) and (2.17), we have

$$-\mu_k \le \langle \bar{G}_\nu^k + \gamma_k(x_k - x_{k-1}), u - x_k \rangle \qquad \forall u \in \mathcal{X}. \tag{B.16}$$

Letting $u = x_*$ in the above inequality and multiplying it by $\alpha_k$, summing it up with (B.15), and denoting $\bar{\Delta}_k = \bar{G}_\nu^k - \nabla f_\nu(w_k)$, we obtain

$$f_\nu(z_k) \le (1-\alpha_k)f_\nu(z_{k-1}) + \alpha_k f_\nu(x_*) + \alpha_k \left[ \mu_k + \langle \bar{\Delta}_k + \gamma_k(x_k - x_{k-1}), x_* - x_k \rangle \right] + \frac{L\alpha_k^2}{2}\|x_k - x_{k-1}\|^2,$$

which together with the facts that

$$\|x_{k-1} - x_*\|^2 = \|x_k - x_{k-1}\|^2 + \|x_k - x_*\|_2^2 + 2\langle x_{k-1} - x_k, x_k - x_* \rangle,$$

$$\alpha_k \langle \bar{\Delta}_k, x_* - x_k \rangle \le \alpha_k \langle \bar{\Delta}_k, x_* - x_{k-1} \rangle + \frac{\|\bar{\Delta}_k\|^2}{2L} + \frac{L\alpha_k^2}{2}\|x_k - x_{k-1}\|^2,$$

imply

$$f_\nu(z_k) \le (1-\alpha_k)f_\nu(z_{k-1}) + \alpha_k f_\nu(x_*) \quad + \quad \alpha_k \left[ \mu_k + \frac{2L\alpha_k - \gamma_k}{2}\|x_k - x_{k-1}\|^2 + \langle \bar{\Delta}_k, x_* - x_{k-1} \rangle \right]$$

$$+ \quad \frac{\alpha_k \gamma_k}{2}\left[\|x_{k-1} - x_*\|^2 - \|x_k - x_*\|^2\right] + \frac{\|\bar{\Delta}_k\|^2}{2L}. \tag{B.17}$$

Defining

$$\hat{\Gamma}_k = \prod_{i=2}^{k}(1 - \alpha_i), \ \ \hat{\Gamma}_1 = 1, \tag{B.18}$$

subtracting $f_\nu(x_*)$ from both sides of the above inequality, diving them by $\hat{\Gamma}_k$, taking expectation, summing them up, noting (A.8) assuming that $\alpha_1 = 1$, $\gamma_k \ge 2L\alpha_k$, and $\gamma_k \alpha_k / \hat{\Gamma}_k$ is constant for any $k \ge 1$, we obtain

$$\frac{\mathbf{E}\left[f_{(z_N)}\right] - f(x_*) - \nu^2 Ld}{\hat{\Gamma}_N} \le \frac{\gamma_1}{2}\|x_0 - x_*\|^2 + \sum_{k=1}^{N}\frac{\alpha_k \mu_k}{\hat{\Gamma}_k} + \left[\frac{(d+5)(B^2 + \sigma^2)}{L} + \frac{\nu^2 L(d+3)^3}{2}\right]\sum_{k=1}^{N}\frac{1}{m_k \hat{\Gamma}_k}.$$

Now noticing that

$$\hat{\Gamma}_k = \frac{2}{k(k+1)}, \qquad \frac{\alpha_k \gamma_k}{\hat{\Gamma}_k} = 4L, \qquad \frac{\alpha_k \mu_k}{\hat{\Gamma}_k} = \frac{LD_0^2}{N},$$

$$\frac{1}{m_k \hat{\Gamma}_k} \le \frac{2D_0^2}{\max\left\{(d+5)B_{L\sigma}N, d+3\right\}}$$

due to (2.19) and (B.18), we obtain (2.20).

Furthermore, note that the function $h_\gamma$ defined in Algorithm 2 is indeed negative the FW-gap of the CG method applied to problem (2.15). From classical analysis of the CG method and similar to our result in Theorem 2.1, one can show that the FW-gap is bounded by $LD_\mathcal{X}^2/T$ if the CG method runs for $T$ iteration. Since the gradient of the objective function in (2.15) is Lipschitz continuous with constant $\gamma$, we have

$$-h_{\gamma_k}(\bar{y}_{T_k}) \le \frac{\gamma_k D_\mathcal{X}^2}{T_k},$$

which together with the choice of $\mu_k$ and $\gamma_k$ in (2.19), imply that at iteration $k$ of Algorithm 1, we need to run Algorithm 2 for at most $T_k = 4D_\mathcal{X}^2 N/D_0^2$ iterations. Therefore, the total number of iterations of Algorithm 2 to find an $\epsilon$-stationary point of problem (1.1) is bounded by $\sum_{k=1}^{N} T_k \le 48LD_\mathcal{X}^2/\epsilon^2$ due to (2.21).

∎

# C   Proofs for Section 3

We now present the proofs for section 3. Recall that, we assumed that $\|\cdot\| = \|\cdot\|_\infty$ in Section 3 We first present two technical results which play key roles in our convergence analysis.

**Lemma C.1** *Let $u \sim N(0, I_d)$ be a d-dimensional standard Gaussian vector. Then for all integer $k \geq 1$ and for some universal constant $C$, we have $\mathbf{E}\left[\|u\|_\infty^k\right] \leq C(2\log d)^{k/2}$.*

*Proof.* Let $Z = \|u\|_\infty$ and denote by $p(x)$ the standard normal pdf. Note that we have

$$\mathbf{E}Z^k = \int_0^\infty kx^{k-1}P(Z > x)\,dx$$
$$\leq \int_0^{x_d} kx^{k-1}dx + \int_{x_d}^\infty x^{k-2}p(x)dx$$

where we define $x_d = \sqrt{2\ln d}$. Now we have

$$\int_0^{x_d} kx^{k-1}dx = x_d^k = (2\log d)^{k/2}$$

and by l'Hospital's rule, for large $d$ we have

$$\int_{x_d}^\infty x^{k-2}p(x)dx \approx x_d^{k-3}p(x_d) \ll (\log d)^{(k-3)/2} = o\left(\frac{(\log d)^{k/2}}{d}\right)$$

Hence we have for some universal constant $C$,

$$\mathbf{E}\left[\|u\|_\infty^k\right] \leq C(2\log d)^{k/2}.$$

■

**Lemma C.2** *The following statements hold for function $f$ and its smooth approximation $f_\nu$.*

   *a)* *Under Assumptions 1 and 2, gradient of $f$ is Lipschitz continuous with constant $L$ and*

$$|f_\nu(x) - f(x)| \leq \nu^2 CL \log d.$$

   *b)* *If Assumption 4 also holds, we have*

$$\|\nabla f_\nu(x) - \nabla f(x)\|_2 \leq C\nu L\sqrt{2s}(\log d)^{3/2}$$
$$\mathbf{E}\left[\|G_\nu(x,\xi;u)\|_\infty^2\right] \leq 4C(\log d)^2\left[L^2\nu^2(\log d) + 4\|\nabla f(x)\|_1^2 + 4\sigma^2\right].$$

*Proof.* First note that

$$|f_\nu(x) - f(x)| = |\mathbf{E}\left[f(x + \nu u) - f(x) - \nu\langle\nabla f(x), u\rangle\right]|$$
$$\leq \mathbf{E}\left|f(x + \nu u) - f(x) - \nu\langle\nabla f(x), u\rangle\right|$$
$$\leq \frac{\nu^2 L}{2}\mathbf{E}\left[\|u\|_\infty^2\right] \leq C\nu^2 L \log d,$$

where the last inequality follows from Lemma C.1. Second, noting this lemma again, Assumption 4, and part a), we have

$$\|\nabla f_\nu(x) - \nabla f(x)\|_2 \leq \sqrt{s^*}\|\nabla f_\nu(x) - \nabla f(x)\|_\infty$$
$$\leq \frac{\sqrt{s}}{\nu(2\pi)^{d/2}}\int |f(x+\nu u) - f(x) - \nu\langle\nabla f(x), u\rangle|\,\|u\|_\infty e^{-\frac{\|u\|_2^2}{2}}\,du$$
$$\leq \frac{\nu L\sqrt{s}}{2(2\pi)^{d/2}}\int \|u\|_\infty^3 e^{-\frac{\|u\|_2^2}{2}}\,du \leq C\nu L\sqrt{2s}(\log d)^{3/2}.$$

Furthermore, by (1.4), Holder inequality, Lemma C.1, and under Assumption 4 we have

$$
\mathbf{E}\left[\|G_\nu(x,\xi;u)\|_\infty^2\right]
$$

$$
= \frac{2}{\nu^2}\mathbf{E}\left[|F(x+\nu u,\xi) - F(x,\xi) - \nu\langle\nabla F(x,\xi),u\rangle|^2\|u\|_\infty^2\right] + 2\mathbf{E}\left[\langle\nabla F(x,\xi),u\rangle^2\|u\|_\infty^2\right]
$$

$$
\leq \frac{\nu^2 L^2}{2}\mathbf{E}\left[\|u\|_\infty^6\right] + 2\mathbf{E}_\xi[\|\nabla F(x,\xi)\|_1^2]\mathbf{E}_u\left[\|u\|_\infty^4\right]
$$

$$
\leq 4CL^2\nu^2(\log d)^3 + 8C(\log d)^2\mathbf{E}_\xi[\|\nabla F(x,\xi)\|_1^2]
$$

$$
\leq 4C(\log d)^2\left[L^2\nu^2(\log d) + 4\|\nabla f(x)\|_1^2 + 4\sigma^2\right].
$$

∎

## C.1 Proof of Theorem 3.1

*Proof.* Noting (1.4), Lemma C.2.a), and with the notion of $G_{\nu,k} \equiv G_\nu(x_k,\xi_k,u_k)$, we have

$$
f(x_{k+1}) \leq f(x_k) + \langle\nabla f(x_k), x_{k+1} - x_k\rangle + \frac{L}{2}\|x_{k+1} - x_k\|_\infty^2
$$

$$
\leq f(x_k) - \gamma_k\langle\nabla f(x_k), G_{\nu,k}\rangle + \frac{L\gamma_k^2}{2}\|G_{\nu,k}\|_\infty^2,
$$

which after taking expectation imply that

$$
\mathbf{E}[f(x_{k+1})] \leq f(x_k) - \gamma_k\|\nabla f(x_k)\|_2^2 + \gamma_k\langle\nabla f(x_k), \nabla f(x_k) - \nabla f_\nu(x_k)\rangle + \frac{L\gamma_k^2}{2}\mathbf{E}[\|G_{\nu,k}\|_\infty^2]
$$

$$
\leq f(x_k) - \frac{\gamma_k}{2}\|\nabla f(x_k)\|_2^2 + \frac{\gamma_k}{2}\|\nabla f(x_k) - \nabla f_\nu(x_k)\|_2^2 + \frac{L\gamma_k^2}{2}\mathbf{E}[\|G_{\nu,k}\|_\infty^2]
$$

$$
\leq f(x_k) - \frac{\gamma_k}{2s}\left(1 - 16LCs(\log d)^2\gamma_k\right)\|\nabla f(x_k)\|_1^2 + (\nu LC)^2 s(\log d)^3\gamma_k
$$

$$
+ 2LC(\log d)^2\left[L^2\nu^2(\log d) + 4\sigma^2\right]\gamma_k^2,
$$

where the last inequality follow from Holder inequality and Lemma C.2.b). Summing both sides of the above inequality over the iterations and rearranging terms, we get

$$
\mathbf{E}[\|\nabla f(x_R)\|_1^2] \leq \frac{6s\left[f(x_0) - f^* + (\nu LC)^2 s(\log d)^3\sum_{k=1}^{N}\gamma_k + 2CL(\log d)^2\left(L^2\nu^2(\log d) + 4\sigma^2\right)\sum_{k=0}^{N-1}\gamma_k^2\right]}{\sum_{k=0}^{N-1}\gamma_k},
$$

where $R$ is uniformly distributed over $\{0,\dots,N-1\}$ since

$$
\mathbf{E}[\|\nabla f(x_R)\|_1^2] = \frac{1}{N}\sum_{k=0}^{N-1}\|\nabla f(x_k)\|_1^2 = \frac{\sum_{k=0}^{N-1}\gamma_k\left(1 - 16LCs(\log d)^2\gamma_k\right)\|\nabla f(x_k)\|_1^2}{\sum_{k=0}^{N-1}\gamma_k\left(1 - 16LCs(\log d)^2\gamma_k\right)},
$$

due to the constant choice of $\gamma_k$ in (3.1). Therefore, we have

$$
\mathbf{E}[\|\nabla f(x_R)\|_1^2] \leq 6s\left[\frac{f(x_0) - f^*}{N\gamma_1} + (\nu LC)^2 s(\log d)^3 + 2CL(\log d)^2\left(L^2\nu^2(\log d) + 4\sigma^2\right)\gamma_1\right],
$$

which together with the choice of smoothing parameter in (3.1) imply (3.2). ∎

## C.2 Proof of Theorem 3.2

*Proof.* Denoting the index set of nonzero elements of $x_k$ and $x_*$ by $Z^k \subseteq \mathbb{R}^{\hat{s}}$ and $Z^* \subseteq \mathbb{R}^{s^*}$, respectively, and $J^k = Z^k \cup Z^{k+1} \cup Z^*$, we have

$$
\|x_{k+1} - x_*\|_2^2
$$

$$
= \|x_{k+1}^{J^k} - x_*^{J^k}\|_2^2 = \|x_k^{J^k} - x_*^{J^k} - \gamma_k G_{\nu,k}^{J^k}\|_2^2 = \|x_k^{J^k} - x_*^{J^k}\|_2^2 + \gamma_k^2\|G_{\nu,k}^{J^k}\|_2^2 - 2\gamma_k\langle x_k^{J^k} - x_*^{J^k}, \gamma_k G_{\nu,k}^{J^k}\rangle
$$

$$
\leq \|x_k - x^*\|_2^2 + (2\hat{s} + s^*)\gamma_k^2\|G_{\nu,k}\|_\infty^2 - 2\gamma_k\langle x_k - x_*, G_{\nu,k}\rangle,
$$

where the inequality follows from the facts that $|J^k| \leq 2\hat{s} + s^*$ and $\|G^{J^k}_{\nu,k}\| \leq \|G_{\nu,k}\|$. Taking expectation from both sides of the above inequality, summing them up, noting Lemma C.2, convexity of $f_\nu$ (due to convexity of $f$), we have

$$\mathbf{E}\left[\|x_N - x^*\|_2^2\right] \leq \|x_0 - x^*\|_2^2 + (2\hat{s} + s^*)\sum_{k=0}^{N-1}\gamma_k^2\mathbf{E}\left[\|G_{\nu,k}\|_\infty^2\right] - 2\sum_{k=0}^{N-1}\gamma_k\langle x_k - x^*, \nabla f_\nu(x_k)\rangle$$

$$\leq \|x_0 - x^*\|_2^2 + 4C(2\hat{s} + s^*)(\log d)^2\sum_{k=0}^{N-1}\gamma_k^2\left[L^2\nu^2(\log d) + 4\|\nabla f(x_k)\|_1^2 + 4\sigma^2\right]$$

$$- 2\sum_{k=0}^{N-1}\gamma_k\left[f_\nu(x_k) - f_\nu(x_*)\right]$$

$$\leq \|x_0 - x^*\|_2^2 + 4C(2\hat{s} + s^*)(\log d)^2\sum_{k=0}^{N-1}\gamma_k^2\left[L^2\nu^2(\log d) + 4\sigma^2\right] + 4\nu^2CL\log d\sum_{k=0}^{N-1}\gamma_k$$

$$- 2\sum_{k=0}^{N-1}\gamma_k[1 - 16LCs(2\hat{s} + s^*)(\log d)^2\gamma_k][f(x_k) - f(x_*)],$$

where the last inequality follows from the fact that $f(x_k) - f(x_*) \geq 1/(2Ls)\|\nabla f(x_k)\|_2^2$ due to the convexity of $f$ and sparsity of its gradient. Rearranging the terms in the above inequality and noting that $\bar{x}_N = \frac{\sum_{k=0}^{N-1}x_k}{N}$, we obtain

$$f(\bar{x}_N) - f(x_*) \leq \frac{\|x_0 - x^*\|_2^2 + 4C(2\hat{s} + s^*)(\log d)^2\sum_{k=0}^{N-1}\gamma_k^2\left[L^2\nu^2(\log d) + 4\sigma^2\right] + 4\nu^2CL\log d\sum_{k=0}^{N-1}\gamma_k}{2\sum_{k=0}^{N-1}\gamma_k[1 - 16LCs(2\hat{s} + s^*)(\log d)^2\gamma_k]}$$

since

$$\bar{x}_N = \frac{\sum_{k=0}^{N-1}x_k}{N} = \frac{\gamma_k[1 - 16LCs(2\hat{s} + s^*)(\log d)^2\gamma_k]x_k}{\sum_{k=0}^{N-1}\gamma_k[1 - 16LCs(2\hat{s} + s^*)(\log d)^2\gamma_k]}$$

due to the constant choice of $\gamma_k$ in (3.5). Hence, (3.6) follows by using the choice of parameters in (3.5) into the above relation. ∎

## D Zeroth-order Stochastic Gradient Method with Inexact Updates-Nonconvex case

In this section, we present a zeroth-order stochastic gradient method which applies the CG method to solve the subproblems. This algorithm shares the main idea of Algorithm 3, but for nonconvex problems. We show while this algorithm enjoys better complexity bound than Algorithm 3, it possess the same one when the same performance measure is employed.

---

**Algorithm 6** Zeroth-order Stochastic Gradient Method with Inexact Updates

Input: $x_0 \in \mathcal{X}$, smoothing parameter $\nu > 0$, positive integer sequence $m_k$, and sequences $\gamma_k$ and $\mu_k$ and a probability distribution $P_R(\cdot)$ over $\{0, \ldots, N-1\}$
**for** $k = 1, \ldots, N$ **do**
    Generate $u_k = [u_{k,1}, \ldots, u_{k,m_k}]$, where $u_{k,j} \sim N(0, I_d)$, call the stochastic oracle $m_k$ times, compute $\bar{G}^k_\nu \equiv \bar{G}_\nu(x_{k-1}, \xi_k, u_k)$ as given by (2.1), and set $x_k$ to (2.17).
**end for**
Output: Generate $R$ according to $P_R(\cdot)$ and output $x_R$.

---

Since we are now using the CG method for inexactly solving (2.15), we can provide an alternative termination criterion than the FW-gap given in (2.5) to provide our convergence analysis. In particular, we use the gradient mapping defined as

$$GP_\mathcal{X}(x, g, \gamma) = \gamma(x - P_\mathcal{X}(x, g, \gamma)), \tag{D.19}$$

where $P_{\mathcal{X}}$ is the solution to (2.15). This quantity which has been widely used in the literature as a convergence criteria for solving nonconvex problems (see, e.g., [24, 25]), plays an analogues role of the gradient in constrained problems. Next result provides some properties for this criteria.

**Lemma D.1** *Let $P_{\mathcal{X}}(\cdot)$ be defined in (2.15), $\gamma > 0$, and $x \in \mathcal{X}$ are given.*

*a) for and $\hat{g} \in \mathbb{R}^d$, we have*

$$\|P_{\mathcal{X}}(x, g, \gamma) - P_{\mathcal{X}}(x, \hat{g}, \gamma)\| \leq \frac{\|g - \hat{g}\|}{\gamma}.$$

*b) Let $P_{\mathcal{X}}^{\mu}$ be the inexact solution of (2.15) such that*

$$\langle g + \gamma(P_{\mathcal{X}}^{\mu}(x, g, \gamma) - x), u - P_{\mathcal{X}}^{\mu}(x, g, \gamma)\rangle \geq -\mu \qquad \forall u \in \mathcal{X} \qquad (\text{D.20})$$

*for some $\mu \geq 0$. Then, we have*

$$\|P_{\mathcal{X}}(x, g, \gamma) - P_{\mathcal{X}}^{\mu}(x, g, \gamma)\|^2 \leq \frac{\mu}{\gamma}.$$

*c) Let $g_x(\cdot)$ be the Frank-Wolfe gap defined in (2.5). Then we have*

$$\|GP_{\mathcal{X}}(x, \nabla f(x), \gamma)\|^2 \leq g_x(x).$$

*Moreover, under Assumption 3, we have*

$$g_x(x) \leq (B/\gamma + D_{\mathcal{X}})\|GP_{\mathcal{X}}(x, \nabla f(x), \gamma)\|.$$

*Proof.* First note that (2.15) implies

$$\|P_{\mathcal{X}}(x, g, \gamma) - P_{\mathcal{X}}(x, \hat{g}, \gamma)\| = \|\Pi_{\mathcal{X}}(x - g/\gamma) - \Pi_{\mathcal{X}}(x - \hat{g}/\gamma)\| \leq \frac{\|g - \hat{g}\|}{\gamma},$$

where the last inequality follows from Lipschitz continuity of the Euclidian projection over the feasible set $\Pi_{\mathcal{X}}$. Second, by optimality condition of (2.15), we have

$$\langle g + \gamma(P_{\mathcal{X}}(x, g, \gamma) - x), u - P_{\mathcal{X}}(x, g, \gamma)\rangle \geq 0 \qquad \forall \tilde{u} \in \mathcal{X}. \qquad (\text{D.21})$$

Letting $\tilde{u} = P_{\mathcal{X}}^{\mu}(x, g, \gamma)$ in the above inequality and $u = P_{\mathcal{X}}(x, g, \gamma)$ and $g = \nabla f(x)$ in (D.20) and summing them up, we clear get the result in part b). Third, letting $\tilde{u} = x$ in (D.21), we have

$$\|GP_{\mathcal{X}}(x, \nabla f(x), \gamma)\|^2 \leq \gamma\langle \nabla f(x), x - P_{\mathcal{X}}(x, \nabla f(x), \gamma)\rangle \leq \gamma g_x(x),$$

where the last inequality follows from (2.5). Furthermore, (D.21) also implies that

$$g_x(x) + \frac{1}{\gamma}\|GP_{\mathcal{X}}(x, \nabla f(x), \gamma)\|^2 \leq \langle \nabla f(x) + \gamma(x - u), x - P_{\mathcal{X}}(x, \nabla f(x), \gamma)\rangle$$

$$\leq (B/\gamma + D_{\mathcal{X}})\|GP_{\mathcal{X}}(x, \nabla f(x), \gamma)\|,$$

where the last inequality follows from Assumption 3.

∎

Now we are ready to state the main result for the nonconvex case.

**Theorem D.1** *Let $\{x_k\}$ be generated by Algorithm 6, the function $f$ be nonconvex, and*

$$\nu = \sqrt{\frac{1}{2N(d+3)^3}}, \quad \gamma_k = 2L, \quad \mu_k = \frac{1}{4N}, \quad m_k = 6(d+5)N, \quad \forall k \geq 1. \qquad (\text{D.22})$$

*Then under Assumptions 1, 2, and 3, we have*

$$\mathbf{E}[\|GP_{\mathcal{X}}(x_R, \nabla f(x_R), \gamma_R)\|^2] \leq \frac{8L}{N}\left(f(x_0) - f^* + L + B^2 + \sigma^2\right). \qquad (\text{D.23})$$

*where $R$ is uniformly distributed over $\{0, \ldots, N-1\}$ and $g_{\mathcal{X}}$ is defined in (D.19). Hence, the total number of calls to the stochastic oracle and linear subproblems solved to find and $\epsilon$-stationary point of problem (1.1) are, respectively, bounded by*

$$\mathcal{O}\left(\frac{d}{\epsilon^2}\right), \quad \mathcal{O}\left(\frac{1}{\epsilon^2}\right). \qquad (\text{D.24})$$

*Proof.* First note that by (1.2), we have

$$f(x_k) \leq f(x_{k-1}) + \langle \nabla f(x_{k-1}), x_k - x_{k-1} \rangle + \frac{L}{2}\|x_k - x_{k-1}\|^2.$$

Letting $u = x_{k-1}$ in (B.16), summing it up with the above inequality, and denoting $\Delta_k = \bar{G}_\nu^k - \nabla f(x_{k-1})$, we obtain

$$f(x_k) \leq f(x_{k-1}) - \gamma_k \left(1 - \frac{L}{2\gamma_k}\right)\|x_k - x_{k-1}\|^2 + \langle \Delta_k, x_{k-1} - x_k \rangle + \mu_k$$

$$\leq f(x_{k-1}) - \gamma_k \left(1 - \frac{L}{\gamma_k}\right)\|x_k - x_{k-1}\|^2 + \frac{\|\Delta_k\|^2}{2L} + \mu_k.$$

Taking expectation from the above inequalities, summing them up, re-arranging the terms, and in the view of Lemma B.1, we have

$$\sum_{k=1}^{N} \gamma_k \left(1 - \frac{L}{\gamma_k}\right) \mathbf{E}[\|x_k - x_{k-1}\|^2]$$

$$\leq f(x_0) - f^* + \sum_{k=1}^{N} \mu_k + \frac{\nu^2 L(d+3)^3 N}{2} + \frac{2(d+5)(B^2+\sigma^2)}{L}\sum_{k=1}^{N}\frac{1}{m_k},$$

which together with the facts that $x_k = P_{\mathcal{X}}^{\mu_k}(x_{k-1}, \bar{G}_\nu^k, \gamma_k)$ and

$$\frac{1}{\gamma_k^2}\|GP_{\mathcal{X}}(x_{k-1}, \nabla f(x_{k-1}), \gamma_k)\|^2$$

$$= \|x_{k-1} - P_{\mathcal{X}}(x_{k-1}, \nabla f(x_{k-1}), \gamma_k)\|^2$$

$$\leq 2\|x_k - x_{k-1}\|^2 + \frac{4\mu_k}{\gamma_k} + \frac{4\nu^2 L^2(d+3)^3}{\gamma_k^2} + \frac{16(d+5)(B^2+\sigma^2)}{\gamma_k^2 m_k},$$

imply that

$$\sum_{k=1}^{N} \left(\frac{\gamma_k - L}{2\gamma_k^2}\right) \mathbf{E}[\|GP_{\mathcal{X}}(x_{k-1}, \nabla f(x_{k-1}), \gamma_k)\|^2] \leq f(x_0) - f^* + \sum_{k=1}^{N}\left(\frac{3\gamma_k - 2L}{\gamma_k}\right)\mu_k$$

$$+ \frac{\nu^2 L(d+3)^3}{2}\sum_{k=1}^{N}\left(1 + \frac{4L(\gamma_k - L)}{\gamma_k^2}\right) + \frac{2(d+5)(B^2+\sigma^2)}{L}\sum_{k=1}^{N}\frac{1}{m_k}\left(1 + \frac{8L(\gamma_k - L)}{\gamma_k^2}\right).$$

Hence, noting (D.22), we obtain

$$\mathbf{E}[\|GP_{\mathcal{X}}(x_R, \nabla f(x_R), \gamma_R)\|^2]$$

$$\leq \frac{8L[f(x_0) - f^*]}{N} + 16L^2\mu_1 + 8\nu^2 L^2(d+3)^3 + \frac{48(d+5)(B^2+\sigma^2)}{m_1},$$

which implies (D.23). Rest of the proof is similar to that of Theorem 2.2 and hence we skip the details. ∎

**Remark 8** *We point out that while the complexity bounds in (D.24) are better than those in (2.9) in terms of dependence on the target accuracy ϵ, they have been obtained for a different performance measure. Indeed, if only the Frank-Wolfe gap is considered then it is easy to see that both bounds are of the same order of magnitude due to part c of Lemma D.1.* ∎