[Reviews · NeurIPS 2018]

Reviewer 1



This paper analyzes a (noisy) zero-order version of the conditional gradient algorithm for both convex and non-convex optimization, and provides bounds on the number of function queries and linear optimization calls that are needed to provide an epsilon-optimal solution, showing that these match the rates of conventional gradient-type algorithms. The algorithm itself is a variant of Nesterov’s smoothing technique (a variant of which dates back to Nemirovsky and Yudin ‘83) used within an outer loop that implements CG with linear optimization calls. They also analyze the “high-dimensional” case where the function only depends on a few of its coordinates, and show improved rates under milder assumptions, showing that there exists come implicit regularization of using conditional gradient updates in this setting. Overall, I liked the paper, and think that the interaction between zero-order optimization and CG updates deserves more attention. The main contribution in terms of algorithm performance seems to be in the high-dimensional regime, and the implicit regularization phenomenon is interesting. The proofs seem more or less correct, but could use some polishing. Given that this is the case, I think the paper meets the quality standard of the conference, and merits publication. A couple of comments: 1. The result of [30] should be added to the table in the introduction. 2. While the table serves most of this purpose, it would be good to have a more detailed comparison with related work, especially in the high-dimensional setting.

Reviewer 2



The authors propose and analyze zeroth-order stochastic approximation algorithms for non-convex and convex optimization. Specifically, they analyze a classical version of CG algorithm and provide results on the convergence rates in the low-dimensional setting; they show a modified algorithm and show that it attains improved rates in the low-dimensional setting; and finally, they consider a zeroth-order stochastic gradient algorithm in the high-dimensional nonconvex (and convex) setting and illustrate an implicit regularization phenomenon. The authors show that this algorithm achieves rates that depend only poly-logarithmically on dimensionality. My detailed comments are as follows: 1. Overall, the paper is very well written and was a pleasure to read. It was clear in its assumptions, definitions and notations and was very easy to follow. 2. The work will be of significance to a lot of readers of the NIPS proceedings especially to the optimization community since it gives an improved algorithm for convex functions in a low-dimensional setting. Moreover, they give an improved bound for the Zeroth-order stochastic Gradient Method for non-convex problems by considering a sparsity assumption. 3. Since this is a theoretical paper, pushing all proofs to the supplementary section makes it a bit hard for the reader. If the authors can give an overview of the proof and point the reader to the details, that would make the paper even better. 4. In section D, the authors use a different termination criterion than the FW-gap. Do earlier results also change with a different termination criteria? Some insights on this would greatly improve the paper. Minor Comments: 1. Algorithms 1 and 3, should also have the input of alpha_k, which is missing.

Reviewer 3



The authors proposed and analyzed zeroth-order stochastic condition gradient and zeroth-order stochastic gradient descent algorithms for nonconvex and convex optimization. They also established the iteration complexities for the proposed algorithms. Pros The zeroth-order stochastic condition gradient algorithm is the first algorithm that explore the stochastic condition gradient by merely using the zeroth-order information of objective. The iteration complexities are well-established. Cons (1) The parameters in the proposed algorithms, such as $\mu_k$ and $m_k$, all depend on the pre-given maximum iteration $N$, which may be hard to estimate an approximated parameter $N$. Why not use adaptive parameters $\mu_k$ and $m_k$ that are independent of the maximum iterations $N$ ? (2) In Theorem 2.1 and 2.2, to reduce the variance and obtain a better iteration complexity rates, the authors require the sample size $mk$ to be $O(N^2)$ and $O(N^3)$, respectively. If $N$ is large, it requires too many function queries. (3) The authors did not provide the iteration complexity of the Algorithm 5(Tuncated ZSGD). Hence, in Table 1, the iteration complexity for Truncated ZSGD should be suitably corrected. (4) In the Algorithm 5, the authors considered an additional projective step onto the sparsity constraint $||x||_0 <= 0$. The iteration complexity for Tuncated ZSGD algorithm is missing. (5) Although the main contributions of this paper are focused on the theoretical iteration complexities, we strongly recommend the authors providing additional experiments to evaluate the efficiency for the new zeroth-order stochastic condition gradient algorithms.

Reviewer 4



Updated comments: I carefully read the authors’ response and the paper again. I was mistaken when I read Algorithms 1 (Eq(2.3)) and 3 (Eq.(2.18)). The authors control the variance by simply averaging a batch of unbiased stochastic gradients. ========================================================= In this paper, the authors considered the problem of zeroth-order (non-)convex stochastic optimization via conditional gradient and gradient methods. However, all the techniques are already known but none are mentioned in the paper. The authors should try to access the recent advances in this topic. In the sequel, the reviewer would like to discuss it in detail. First, the main trick that makes Algorithm 1 works is Eq. (2.3) (and (2.18) for Algorithm 3). To the best of the reviewer’s knowledge, this technique was first applied to the conditional gradient method and stochastic optimization in the following line of work Centralized setting: [1] Mokhtari, Aryan, Hamed Hassani, and Amin Karbasi. "Conditional Gradient Method for Stochastic Submodular Maximization: Closing the Gap." arXiv preprint arXiv:1711.01660 (2017). [2] Mokhtari, Aryan, Hamed Hassani, and Amin Karbasi. "Stochastic Conditional Gradient Methods: From Convex Minimization to Submodular Maximization." arXiv preprint arXiv:1804.09554 (2018). Specifically, in [2], Mokhtari et al. applied the averaging technique to conditional gradient method for stochastic optimization, which is exactly the same setting and almost the same algorithm. Even the proof technique is essentially the same (page 13 of this paper). Their method with the unbiased estimator in the equation right below line 96 works in this zeroth-order setting because the unbiased estimator naturally provides randomness in the gradient oracle. Both Mokhtari et al.’s algorithm and the proposed algorithm has an O(1/\epsilon^3) rate (to be precise, the rate of Algorithm is O(d/\epsilon^3), where the linear dependence on d is due to the unbiased estimator). However, the authors claimed “But the performance of the CG algorithm under the 36 zeroth-order oracle is unexplored in the literature to the best of our knowledge, both under convex and 37 nonconvex settings.” In addition, Mokhtari et al.’s algorithm only needs one sample per iteration while the proposed algorithm (Algorithm 1) needs a batch of several samples, which is even worse. Algorithm 2 is basically like a proximal Nesterov-type acceleration. Again, the main trick in Algorithm 3 is the averaging (2.18). The authors should do extensive literature review and compare their results with existing work. This paper, in its current form, cannot be accepted without proper comparison with existing work.

Reviewer 5



The paper considers the problem of extending the Frank-Wolfe method to scenarios that only an unbiased estimate of the function value is available, i.e. we don't have access to any (stochastic) gradient o higher order information and only (an unbiased version of) the function values can be obtained from the oracle. The authors use Nestrov's smoothing method to find a (stochastic) proxy of the gradient and use such a proxy to design their algorithms. I have carefully read the paper and I have two major issues with this paper. While I find the problem and the solutions interesting and somehow novel, I believe the following two issues have to be resolved before I can give a higher mark for accepting the paper. 1) Related work and giving credit to previous work: Algorithm 1 in the paper is exactly the algorithm SFW proposed in [12,24] with the difference that the stochastic gradients are replaced by unbiased estimators of the smoothed function (from equation (1.3)). While I find the smoothing step crucial and novel due to having only zero-order information, I believe that the authors should have given credit to [12] or [24] by specifying what steps Algorithm 1 has different with respect to those algorithms. Furthermore, the problem of stochastic FW (with first order information) has been studied recently in various papers and the authors have failed to review recent progress and related work on this topic. For example, the following paper by Mokhtari et al proposes a stochastic Frank-Wolfe method for both convex and structured non-convex problems with O(1/eps^3) samples (potentially dimension-independent): https://arxiv.org/abs/1804.09554 I expect that in the revised version the authors highlight the differences of their algorithms with the recent results (perhaps the authors can add a section called related work). I understand that the authors consider zero-order information, but in principle by using the smoothing trick one can come up with (unbiased estimates of) first order information of a sufficiently close function (as done in equation 1.3 in the paper). 2) To get the O(d/eps^2) bound the authors add a proximal step (equation 2.15) and solve this step by using the FW method (Algorithm 2). Well, I can not really call the whole algorithm (Algorithm 3) a stochastic and projection-free conditional gradient method. The authors are just solving an expensive quadratic optimization problem by the FW method and claim that the overall algorithm is a FW-type method. Indeed, one can solve any quadratic projection problem with FW. For example, consider projected gradient descent. We can solve the projection step with FW and claim that we have a projection-free method with rate O(1/eps^2). Can the authors comment on this?